

# Research on the relationship and prediction model between nighttime lighting data, pm2.5 data, and urban GDP

Sen Chen and Junke Li

School of Information Engineering, Suqian University, Suqian, Jiangsu, China

## ABSTRACT

With the discovery of electricity and the widespread adoption of lighting technology, the extensive application of electricity has greatly increased productivity, making night-time factory production possible. At the same time, the rapid expansion of factories has led to a significant increase in particulate matter 2.5 (PM2.5) in the air. However, economic development heavily relies on lighting and factory production. To address this issue, researchers have focused on predicting urban gross domestic product (GDP) through night-time lights and PM2.5, but current studies often focus on the impact of a single factor on GDP, leaving room for improvement in model accuracy. In response to this problem, this article proposes the Relationship and Prediction Model between Night Light Data, PM2.5, and Urban GDP (R&P-NLPG model). Firstly, night light data, PM2.5 data, and GDP data are collected and preprocessed. Secondly, correlation analysis is conducted to analyze the correlation between data features. Then, data fusion methods are used to integrate features between night-time data and PM2.5 data, forming the third data features. Next, a neural network is constructed to establish a functional relationship between features and GDP. Finally, the trained neural network model is used to predict GDP. The experimental results demonstrate that the predictive capability of the R&P-NLPG model outperforms GDP prediction models constructed with single-feature input and existing multi-feature input.

## INTRODUCTION

With the development of electricity, artificial lighting has rapidly evolved, altering the traditional lifestyle of working with the sun and resting with the moon. It has met the needs of human nighttime activities. Industrial and domestic electricity usage has greatly improved living conditions, and the application of lighting electronic technology in daily life has profoundly changed and influenced people's modes of production (*Liu, 2023*). Due to the demand for production materials, many factories in various places are brightly lit at night. Nowadays, many factories adopt a "three-shift" work model, continuing production at night. Factory production has become more efficient at night, greatly promoting the development of urban gross domestic product (GDP). To grasp the distribution of brightness on the Earth's surface caused by human nighttime activities, the United States

Corresponding author
Junke Li, junker_li@squ.edu.cn

initiated the Defense Meteorological Satellite Program in the 1970s. It aims to capture the faint light radiation from the surface at night and produces a series of annual cloud-free nighttime light images. Through the analysis of nighttime images, there is a clear correlation between the intensity of nighttime lighting and the level of urban development (*Qin, 2021*). Therefore, the development of regional economies is inseparable from lighting, and nighttime lighting data can better reflect the local GDP situation (*Zhu & Song, 2025*). It can be speculated that there is a certain relationship between the level of nighttime lighting in a region and the level of urban GDP in that region.

Driven by the first Industrial Revolution, with the advent of the steam engine, human productivity was greatly increased. This allowed factories and manufacturing to produce goods more efficiently, thus promoting the process of industrialization and the development of the social economy. However, the invention of the steam engine also led to a significant increase in the concentration of particulate matter 2.5 (PM2.5) in the air (*Yang, Fan & Zhao, 2020*). PM2.5 refers to solid or liquid particulate matter with a diameter of 2.5 micrometers or less in environmental air. It is a complex and variable atmospheric pollutant composed of a large number of different chemical components, which produced by both artificial and natural pollution sources and can be suspended in the air. PM2.5 is an atmospheric pollutant generated from both anthropogenic and natural sources. In 2012, *Chen (2025)* pointed out that PM2.5 pollution caused economic losses amounting to 6.82 billion yuan according to epidemiological studies. In the Beijing-Tianjin-Hebei region, losses reached 172.9 billion yuan in 2009 and 134.29 billion yuan in 2013 (*Chen, 2025*). The Yangtze River Delta region recorded losses of 22.1 billion yuan in 2010 (*Chen, 2025*). In the Beijing-Tianjin-Hebei region, the losses due to PM2.5 pollution were 172.9 billion yuan in 2009 and 134.29 billion yuan in 2013, and in the Yangtze River Delta region, it caused a loss of 22.1 billion yuan in 2010 (*Chen, 2025*). From the above data, it can be seen that the overall situation of PM2.5 pollution in China is severe. Severe smog pollution not only harms the health of residents but also leads to huge economic losses and hinders further economic development (*Cao, 2020*).

While both factors can reflect urban GDP development, existing researches primarily rely on single-factor models for GDP prediction and analysis, overlooking the accuracy limitations of univariate approaches. This article presents a GDP prediction model incorporating multiple factors *via* feature indicators.

The primary contributions of this study are outlined as follows:

(1) current GDP prediction models predominantly rely on univariate approaches, largely neglecting the multifaceted determinants of urban development. To address this limitation, this study proposes a novel multivariate framework through integrated feature analysis.

(2) We propose the Relationship and Prediction Model between Night Light Data, PM2.5, and Urban GDP (R&P-NLPG) model. This model innovatively employs convex combination to integrate two feature variables (nighttime light and PM2.5), generating a third composite feature for GDP prediction. Subsequently, the neural network is

trained using historical data to obtain optimal network weights, ultimately yielding the predictive model.

(3) We employed mean squared error (MSE) and the correlation coefficient (R-value) to evaluate the accuracy of the model. For assessing the model's robustness, we implemented the Mann-Whitney U hypothesis testing to demonstrate the model's stability. The results show that the proposed model is effective and exhibits high accuracy.

## RELATED WORKS

In recent years, a significant number of researchers have been involved in studying the prediction of GDP using nighttime light data and PM2.5. The following outlines the current state of relationship analysis and predictive modeling between nighttime light data and PM2.5 in relation to GDP.

### Analysis and prediction model of the relationship between nighttime lighting imagery and GDP

Nighttime light data can effectively reflect the intensity of human nighttime activities. Drawing on recent advances in the use of remote sensing data to estimate economic activities, *McSharry & Mawejje (2024)* identify the observable variables that are correlated with nighttime lights in urban South Sudan. By using the machine learning aided regression model to forecast future nighttime lights, they convert it into an estimate of the GDP growth (*McSharry & Mawejje, 2024*). The study of *Huang et al. (2021)* confirmed that the Location-Based Social Media-Technische Universität Dresden (LBSM-TUD) data are a potential and promising data source for economic modeling in small scale areas of China, which will help to support China's regional economic evaluation. Due to the lack of annual Suomi National Polar-orbiting Partnership-Visible Infrared Imaging Radiometer Suite (NPP-VIIRS) nighttime light data, *Li & Tan (2023)* took Yunnan Province as an example to calibrate the NPP-VIIRS imagery on a monthly basis. The experimental results showed that the correlation between the synthesized annual nighttime light data and GDP reached 0.996, providing a favorable basis for our analysis of the relationship between nighttime lights and GDP (*Li & Tan, 2023*). However, they only consider the light intensity of nighttime construction land, and the lights data from many commercial areas in cities are also significant and should not be overlooked. The satellite nighttime light images were also used to identify the relationship between urbanization and road networks by *Kulpanich et al. (2023)*. Analysis of these data indicated that urbanization and road networks were significantly positively correlated at the 0.01 level (*Kulpanich et al., 2023*). *Zhan, Hu & Liu (2021)* confirmed the linear relationship between GDP and Suomi National Polar-orbiting Partnership Visible Infrared Imaging Radiometer Suite (Suomi NPP-VIIRS) nighttime light data in GDP prediction model. His research incorporate nighttime light data as a feature in GDP models. *Rao, Wang & Li (2024)* calibrated the historical GDP statistical data of the Pearl River with new "NPP-VIIRS-like" nighttime

light data and constructed two best economic growth simulation scenarios based on the carbon emission reduction rate.

However, *Yu & Gao (2022)* found that Defense Meteorological Satellite Program's Operational Linescan System (DMSP/OLS) nighttime light data have the characteristic of a long time series and it has certain advantages over NPP-VIIRS nighttime light data. Subsequently, *Jia & Qin (2021)* constructed a GDP prediction model based on DMSP/OLS nighttime light data to estimate and analyze the GDP values of three cities. The results showed that there is a linear correlation between nighttime light data and GDP values (*Jia & Qin, 2021*). *Chang et al. (2022)* combined three nighttime light indices with GDP data to fit a regression model for 18 cities and counties in Hainan Island from 2012 to 2018. The results showed that the three light indices had the best linear correlation with GDP.

The above study used different methods to investigate the correlation between nighttime lighting and GDP, and the results showed significant correlations in different regions.

## Analysis and prediction model of the relationship between PM2.5 and GDP

A study found that industrial trends are closely correlated with industrial transformation/upgrading and regional policies. This development often accompanies air pollutant emissions (*Xiong & An, 2020*). *Shi et al. (2019)* used spatial regression model to quantify the impact of urban form on PM2.5 concentration. The results showed that urban form indicators are significantly correlated with PM2.5 concentration. *Zhang, Zhang & Hu (2020)* adopted a spatial variable coefficient model to reveal that the increase in energy consumption per unit of GDP are positively correlated with smog pollution. The reasons include the blind promotion of urbanization construction without considering the urban environmental capacity and whether the urban population size is reasonable (*Zhang, Zhang & Hu, 2020*). *Mandal & Thakur (2023)* proposed a PM2.5 concentration prediction model based on spatial attention clustering (SA-GNN). This model predicted short-term PM2.5 concentrations by considering monitoring stations as nodes in the graph structure and exploring their spatial relationships (*Mandal & Thakur, 2023*). *Li et al. (2025)* proposed a short-term PM2.5 concentration prediction framework based on the SA-GNN model which can be applicable to the heavily polluted Indian capital, Delhi. The proposed model achieved better performance compared with the baseline model on the test data (*Li et al., 2025*).

Based on existing PM2.5 research methods, *Cao (2016)* demonstrated the correlation between GDP spatial distribution and PM2.5 concentration through nighttime light imagery. Economic development levels demonstrate significant correlation with industrialization progress. Higher industrialization levels indicate greater urban development levels and GDP in corresponding regions (*Ren, 2022*). Therefore, it can be concluded that the PM2.5 content in a region is inextricably linked to the industrialization level of that region.

Current research has confirmed the correlations between nighttime light data and GDP, as well as PM2.5 and GDP. While existing models for GDP prediction are well-developed, they predominantly rely on single variable, overlooking the multifaceted nature of urban development. To improve the accuracy of urban GDP forecasting, this study proposes the R&P-NLPG model, which integrates multiple factors *via* a convex combination approach.

## R&P-NLPG MODEL

The R&P-NLPG model process is shown in Fig. 1. Initially, nighttime light data, PM2.5 data, and GDP data are collected and normalized to form preliminary data. Subsequently, the Pearson correlation is used to verify the correlation between nighttime light imagery, PM2.5 and GDP. Then, through data fusion, the nighttime data and PM2.5 data are merged as third feature indicators. Then, the three features are input into the neural network for function fitting. Finally, the trained fitting function is used to predict GDP.

### Data collection and preprocessing

#### *Nighttime lighting data collection*

The nighttime light data used in the study is provided by the Earth observation Group of the National Centers for Environmental Information (NCEI), a part of the National Oceanic and Atmospheric Administration (NOAA) in the United States. The data adopts the World Geodetic System 1984 (WGS 1984) geographic coordinate system, offering a global surface average nighttime light value for each year from 2012 to 2021. Compared to the original products, the data provided by Earth observation Group has higher accuracy, stability, and ease of use. It can be utilized to continuously track human activities on the Earth's surface. The version of the data used in this article has units of nanowatts per square centimeter per steradian ($nW/cm^2/sr$).

The specific process of extracting nighttime light data is shown in Fig. 2. First, obtain the nighttime light data map, then use the Spatial Analyst tool in the ArcGIS application to perform data cropping, projection, and resampling for data calculation. The data is processed using provincial administrative units as the raster data to extract the nighttime light values. Finally, the extracted data is exported.

#### *PM2.5 data collection*

The process for obtaining PM2.5 data is shown in Fig. 3. Firstly, select the annual average PM2.5 concentration images extracted from the Atmospheric Composition Analysis Group at Dalhousie University. Then, the image data is inverted through the Moderate Resolution Imaging Spectroradiometer (NASA MODIS), Multi-angle Imaging Spectro Radiometer (MISR), Sea-viewing Wide Field-of-view Sensor (Sea WIFS), and Aerosol Optical Depth (AOD) tools. By combining the above inverted results with the GEOS-Chem chemical transport model and using geographically weighted regression (GWR) to correct the regional observation data, the annual average PM2.5 concentration image data was finally generated.

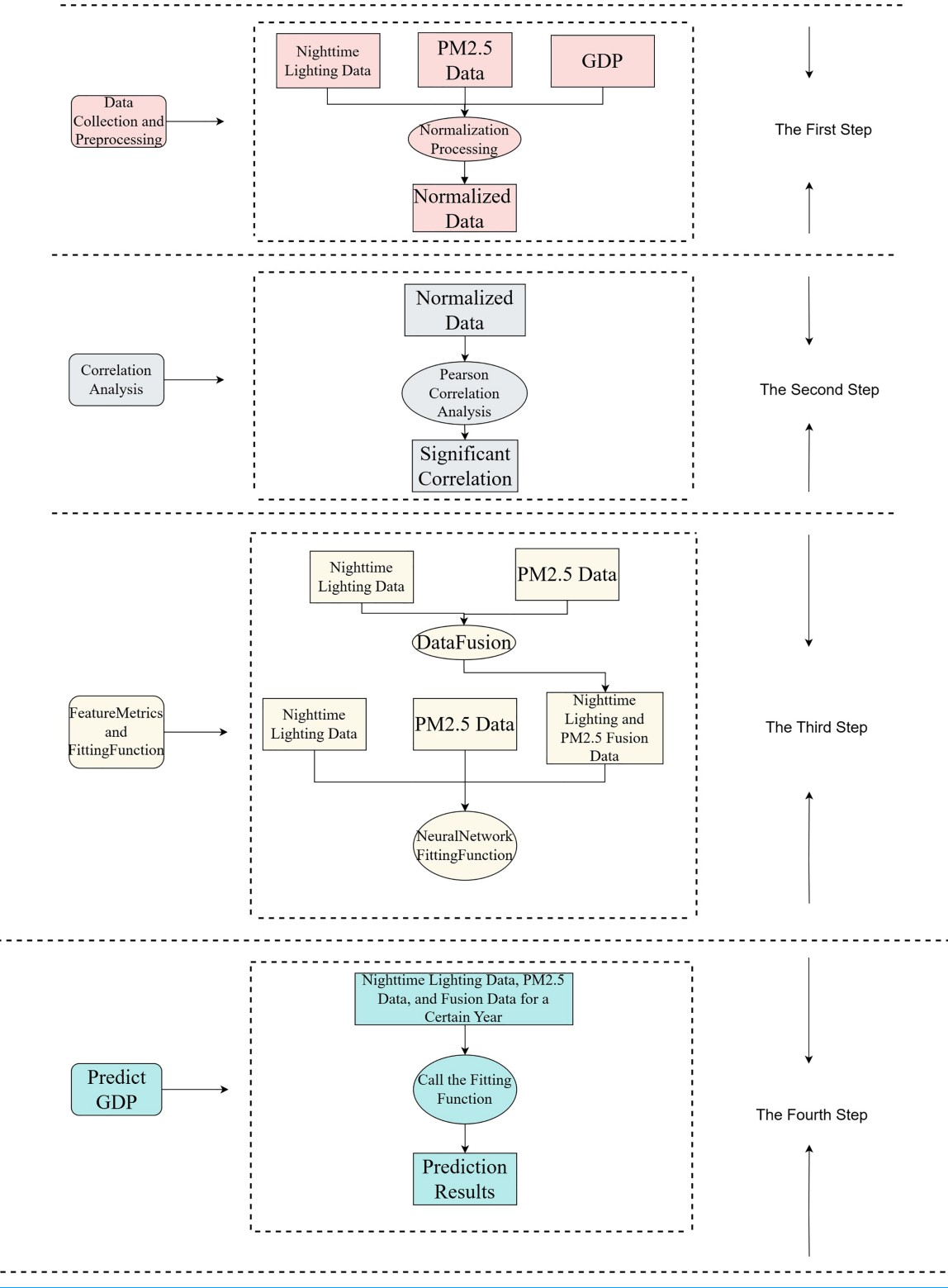

**Figure 1  R&P-NLPG model framework.**

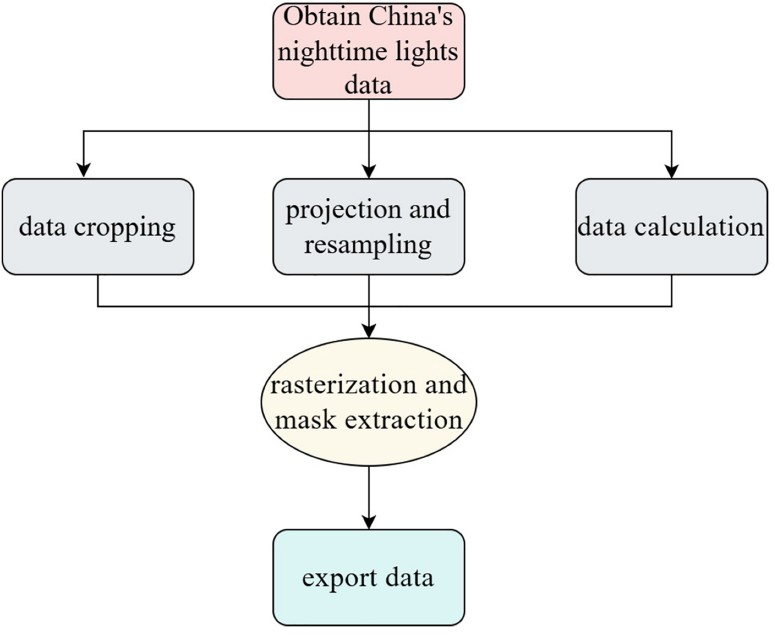

**Figure 2 Nighttime lighting acquisition process.**

### GDP data collection

The annual GDP data for each province in China comes from the (China Statistical Yearbook from 2012 to 2021), selecting 34 provincial-level administrative regions in China (excluding Hong Kong, Macao, and Taiwan).

### Data preprocessing

In the R&P-NLPG model, due to the different magnitudes of the two-types input data and the different evaluation criteria for sample data, it is necessary to standardize the evaluation criteria to prevent data with small values from being overwhelmed. When the value of a data item is very large, direct calculation may exceed the value range of common data types. Logarithmic transformation can reduce the absolute value of the data, facilitating subsequent calculations and processing. Therefore, logarithmic processing is required for the data. The specific logarithmic processing formula is shown in Eq. (1).

$$y = \ln(x). \tag{1}$$

### Correlation analysis

We use Pearson correlation coefficient to get the correlation between PM2.5 data, nighttime light data and GDP data. It can be calculated by Eq. (2).

$$rxy = \frac{n\sum_{i=1}^{n}X_iY_i - \sum_{i=1}^{n}X_i\sum_{i=1}^{n}Y_i}{\sqrt{n\sum_{i=1}^{n}X_i^2 - \left(\sum_{i=1}^{n}X_i\right)^2}\sqrt{n\sum_{i=1}^{n}Y_i^2 - \left(\sum_{i=1}^{n}Y_i\right)^2}}. \tag{2}$$

In Eq. (2), the nighttime light data and PM2.5 data respectively serves as the independent variable $x$, and the GDP data serves as the dependent variable $y$. $rxy$ is the

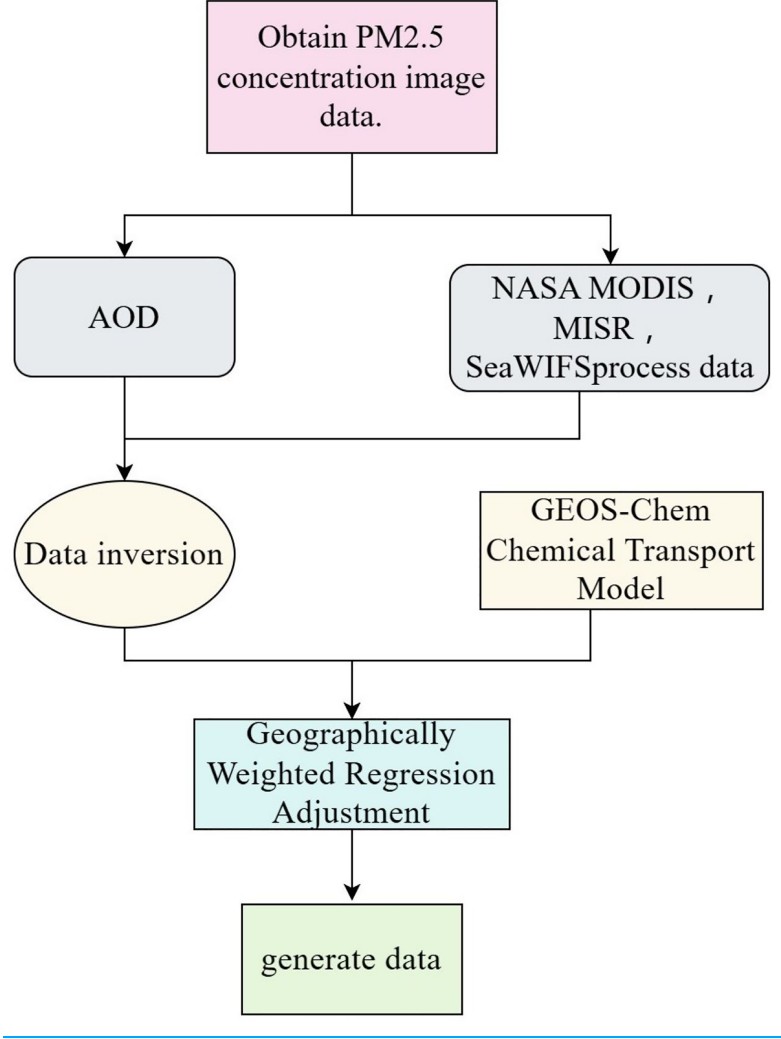

**Figure 3  PM2.5 data acquisition process.**

Pearson correlation coefficient between variables $x$ and $y$; $n$ is the number of observations; $Xi$ is the $i$-$th$ observation of $x$; $Yi$ is the $i$-$th$ observation of $y$.

## Feature fusion and fitting function

### Feature fusion

There are three input features in the R&P-NLPG model, namely PM2.5, nighttime light, and the fused feature after the convex combination of PM2.5 and nighttime light. The convex combination method for GDP and PM2.5 values is shown in Eq. (3).

$$Z = \theta x + (1 - \theta)y$$
$$0 < \theta < 1. \tag{3}$$

In the formula, $x$ represents the nighttime light data, $y$ represents the PM2.5 data, and $Z$ represents the convex combination of nighttime light and PM2.5 data. The value of $\theta$ is

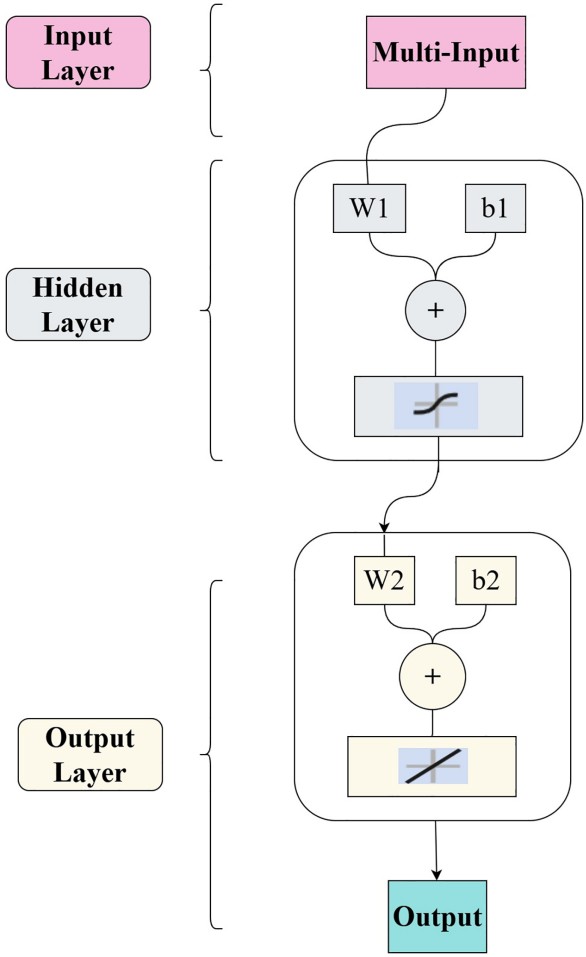

**Figure 4  BP neural network structure.**                 

obtained through experiments, which will be described in detail in the experimental section.

### *Fitting function*

The R&P-NLPG model employs a backpropagation (BP) neural network as the fitting function. The experimental BP architecture includes one input layer, one hidden layer and one output layer. Connections between layers are represented by weights. Each node in the BP network comprises a perceptron (a single neuron) with inputs, weight, biases, activation functions (Sigmoid and Rectified Linear Unit (ReLu)), and outputs. To mitigate potential overfitting, the Bayesian regularization algorithm was applied for neural network training. Additionally, *k*-fold cross-validation is also applied during the training. The BP architecture within the R&P-NLPG model is structured as illustrated in Fig. 4.

During the forward propagation process, the input data is calculated through the perceptron nodes and then processed by the activation function to obtain the output results. In the backward propagation process, the results are compared with the expected

**Table 1 Nighttime lights data.**

| Province | Nighttime light data (nW/cm²/sr) | Province | Nighttime light data (nW/cm²/sr) |
|---|---|---|---|
| Anhu | 1.691745235 | Liaoning | 1.390142897 |
| Beijing | 5.856288846 | Inner Mongolia | 0.45853 |
| Fujian | 1.846049842 | Ningxia | 1.128802 |
| Gansu | 0.491204472 | Qinghai | 0.385857 |
| Guangdong | 3.07086353 | Shandong | 1.202445 |
| Guangxi | 0.757481083 | Shanxi | 2.524983 |
| Guizhou | 0.675371318 | Shaanxi | 1.02245 |
| Hainan | 1.505900505 | Shanghai | 14.59642 |
| Hebei | 1.473290594 | Sichuan | 0.629282 |
| Henan | 1.711040447 | Tianjin | 7.557534 |
| Heilongjiang | 0.622681397 | Tibet | 0.356043 |
| Hubei | 0.986916101 | Xinjiang | 0.489051 |
| Hunan | 0.769430431 | Yunnan | 0.589511 |
| Jilin | 0.867317229 | Zhejiang | 3.526563 |
| Jiangsu | 4.466510869 | Chongqing | 1.177244 |
| Jiangxi | 0.890769623 | | |

results, and the weights of the nodes in the network are continuously adjusted through multiple iterations. The calculation method for each layer is represented by Eq. (4).

$$y = T(Wx + b). \tag{4}$$

In Eq. (4), $T$ represents the activation function, $W$ represents the weights (the higher the weight, the more important the feature is), and $b$ represents the activation threshold (bias term).

To address potential overfitting, this experiment applied $k$-fold cross-validation (with $k = 1$, *i.e.*, leave-one-out cross-validation) as follows:

Step 1: randomly split the original data into $k$ non-overlapping subsets.

Step 2: use $k$-1 subsets for model training and the remaining subset for testing.

Step 3: repeat Step 2 $k$ times to obtain $k$ models and their evaluation results.

Step 4: calculate the average cross-validation score for model performance assessment.

## RESULTS

### Data acquisition and preprocessing

The nighttime light data in China will be compiled annually using the component arcmap in ArcGIS. Table 1 shows the extracted partial nighttime light data.

PM2.5 data is extracted from the image data provided by the Atmospheric Composition Analysis Group at Dalhousie University in Canada. GDP data is directly retrieved from the China Statistical Yearbook. After that, these data are processed using Eq. (1) to obtain preliminary experimental data.

**Table 2 Pearson correlation coefficient.**

|  | GDP | PM2.5 | Lighting |
|---|---|---|---|
| GDP | 1 |  |  |
| PM2.5 | 0.483** | 1 |  |
| Lighting | 0.433** | 0.419** | 1 |

Note:
** At the 0.01 level (two-tailed), the correlation is significant.

## Correlation analysis

The total nighttime light data, total PM2.5 concentration data, and GDP data for 31 provinces from 2012 to 2021 were analyzed using SPSS software to obtain the correlation coefficient.

The results of the correlation analysis are shown in Table 2. The analysis shows that PM2.5 is significantly correlated with GDP at the 0.01 level (two-tailed). The nighttime light data is significantly correlated with GDP at the 0.01 level (two-tailed). The nighttime light data is significantly correlated with PM2.5 at the 0.01 level (two-tailed).

## Fitting results

In the experiment, the BP structure is set as follows: the input layer, the hidden layer and the output layer respectively have three neurons and 10 neurons with the tansig activation function, and one neuron with the ReLU activation function. For the weight between input layer and the hidden layer, it is represented by a $10 \times 3$ matrix $W1$, and the biases by a $10 \times 1$ vector $b1$. For the weight between the hidden layer and output layer, the weights are a $1 \times 10$ matrix $W2$, and the bias is a scalar $b2$. To get best fitting, the model was trained 10 times. Through cross-validation, the weight matrices and bias matrices for the BP neural network were ultimately derived as follows:

$$W1 = \begin{bmatrix} 1.08169 & 2.39017 & -1.43835 \\ 2.21875 & -1.92324 & 3.41236 \\ 0.31239 & 2.96546 & -1.11123 \\ 0.43255 & 3.00542 & 3.60995 \\ -2.23039 & -1.06701 & -0.93613 \\ 4.41689 & -2.91412 & 2.77755 \\ 2.83761 & -3.95985 & -5.54481 \\ -2.55980 & 1.31360 & 1.08317 \\ 1.13775 & -2.02131 & -2,39326 \\ 2.76203 & 0.54822 & 1.73565 \end{bmatrix} \quad b1 = \begin{bmatrix} 0.0486 \\ 1.5192 \\ 0.0022 \\ -0.8724 \\ -0.5954 \\ -0.7349 \\ -2.3825 \\ 3.9511 \\ -0.3711 \\ 7.3459 \end{bmatrix}$$

$$W2 = \begin{bmatrix} 0.15 & 1.32 & -1.18 & -3.02 & 1.46 & -1.32 & -2.54 & -0.43 & -1.39 & 2.13 \end{bmatrix}$$

$$b2 = \begin{bmatrix} -0.81795 \end{bmatrix}.$$

We set the convex combination coefficient $\theta$ as 0.5, and then train the BP neural network. Figure 5 presents the mean squared error (MSE) progression during neural network training. The horizontal axis denotes training epochs (0–11), while the vertical axis represents MSE values. At each epoch, weights trained on the training set were also validated using the test set. The optimal validation performance (MSE = 0.014672) was
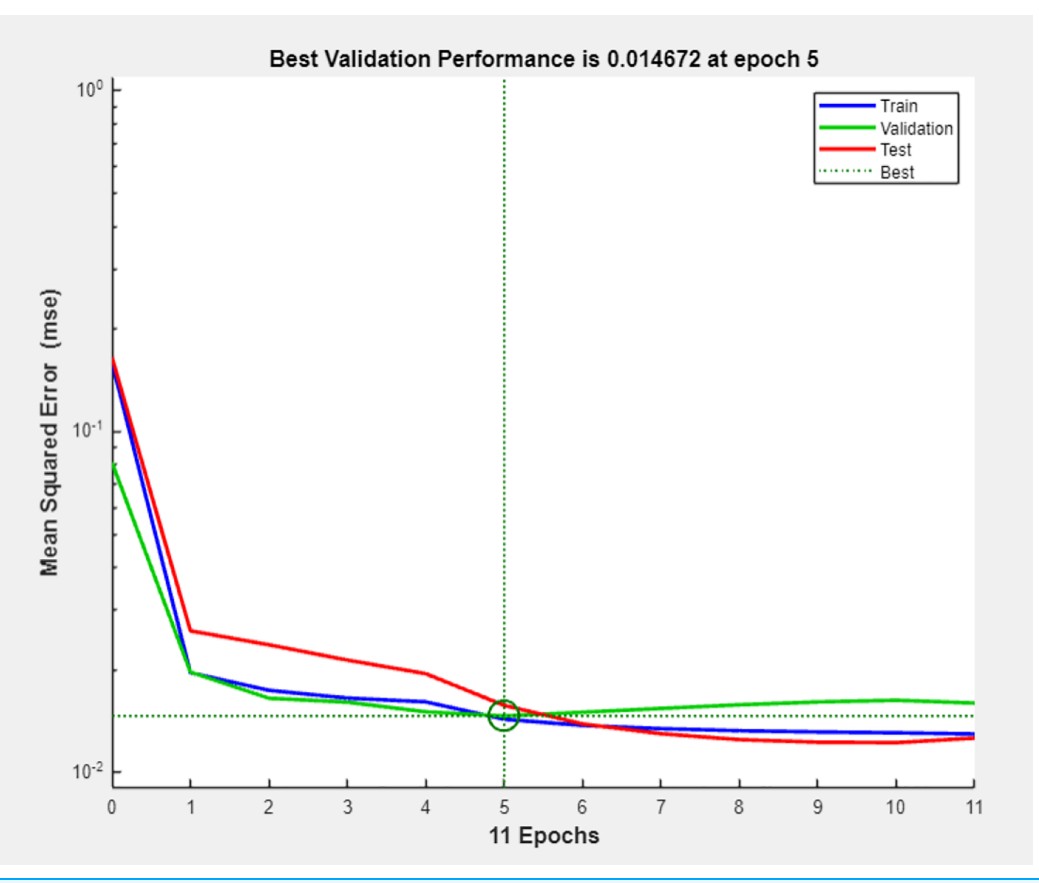

**Figure 5** **Mean squared error of training and test sets.**

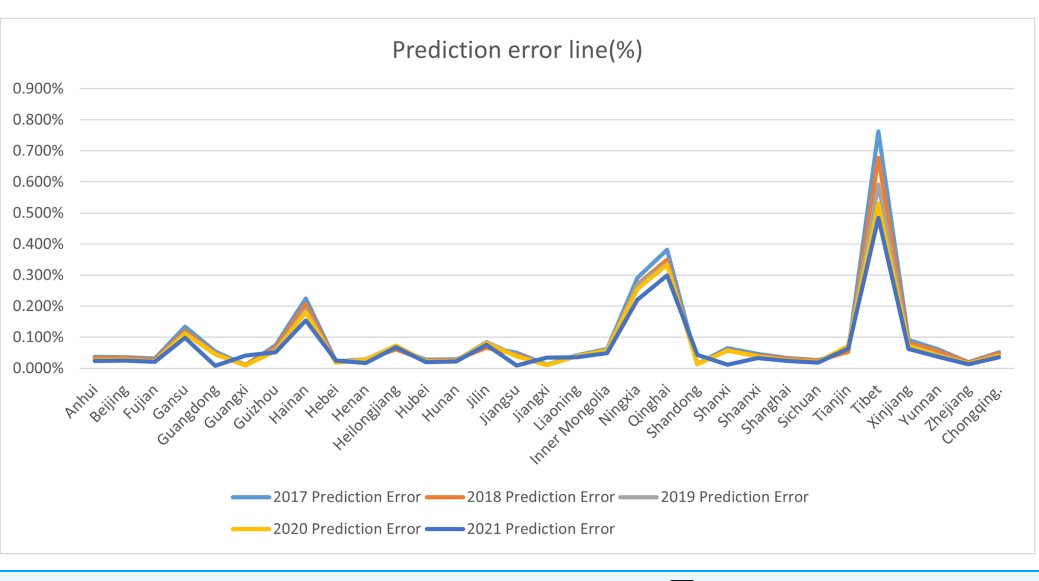

**Figure 6** **Prediction error line (%).**

Table 3  Mann-Whitney U hypothesis testing.

| Group | Median difference and 95% CI | Rank-sum test | |
|---|---|---|---|
| | | Z value | p value |
| Actual GDP group R&P-NLPG-predicted GDP group | 3,223.624 [103.373–6,989.039] | 2.035 | 0.668 |

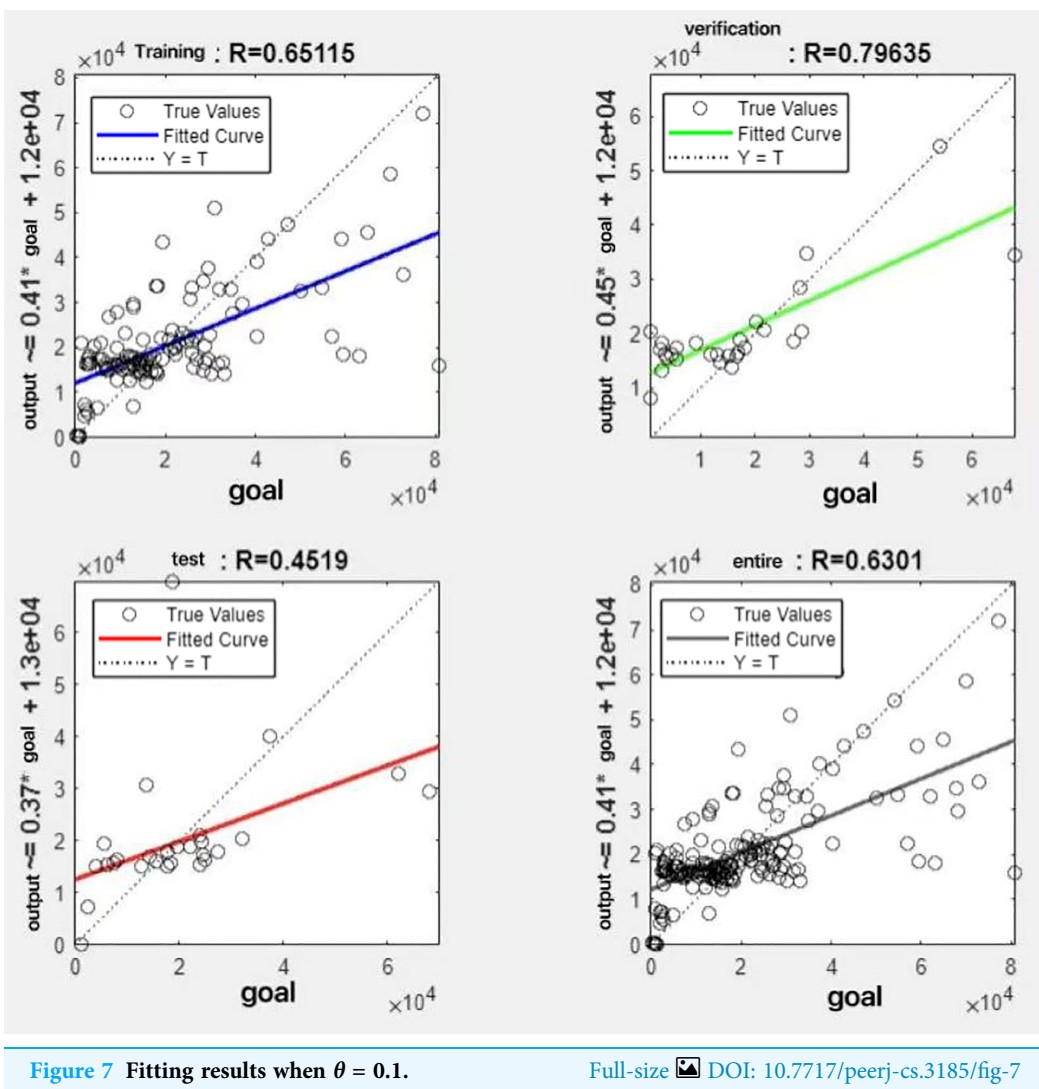

**Figure 7 Fitting results when θ = 0.1.**

observed at epoch 5, beyond which the overfitting occurred. Consequently, epoch 5 was identified as the optimal checkpoint. The mean square error is shown in Fig. 5.

To evaluate the model's accuracy, we use the Eq. (5) to calculate the error. These details are shown in the Fig. 6.

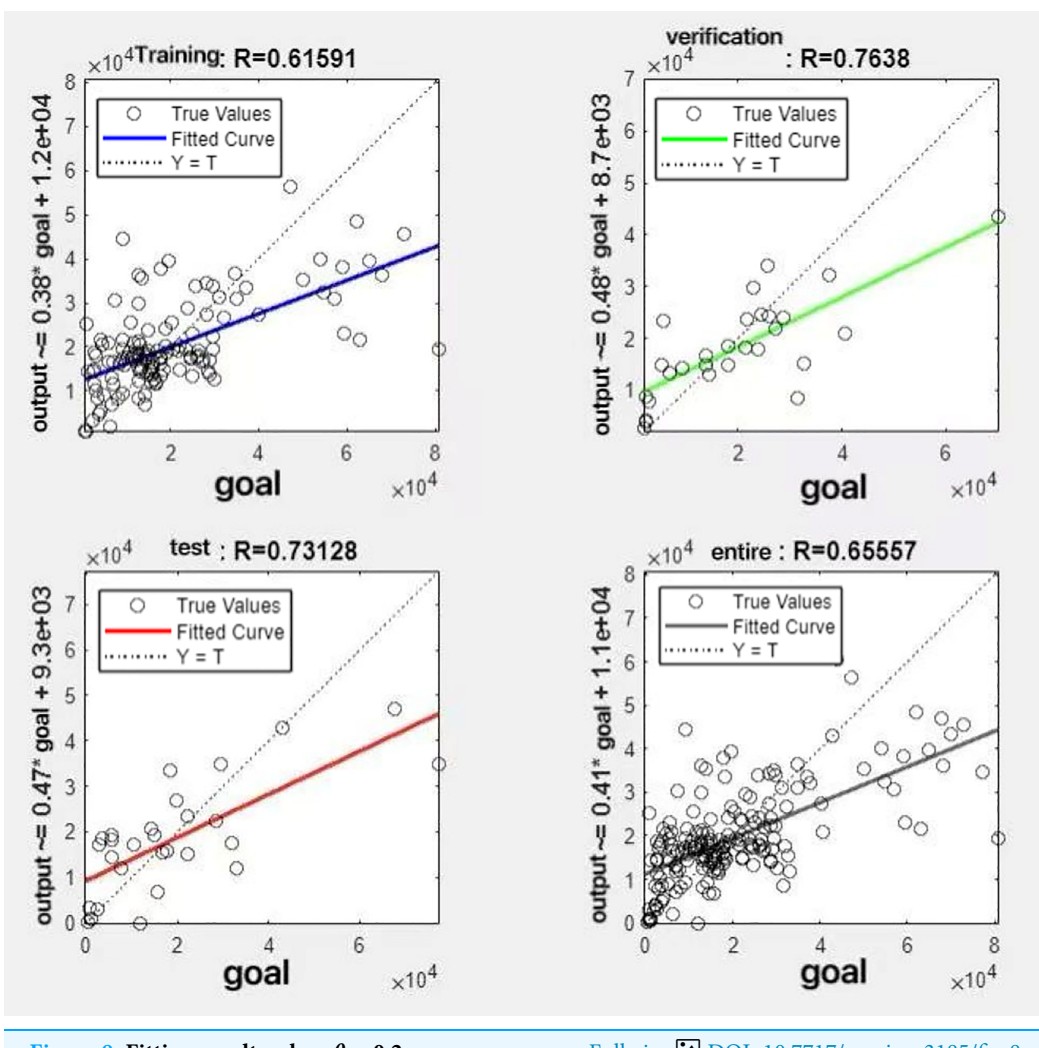

**Figure 8** Fitting results when θ = 0.2. 

$$error = \frac{|actual\ GDP\ value - predicted\ value|}{actual\ GDP\ values} \times 100\%. \tag{5}$$

From Fig. 6, we can see that the forecast error ranged between 0% and 0.8% in 2021. The errors of other predicted years follow a similar trend and are omitted for brevity. The results demonstrate that the error between our predicted values and the true values is minimal, meeting practical requirements.

## Stability of the model

To rigorously evaluate the stability of the proposed model, a Mann-Whitney U hypothesis testing was systematically deployed to statistically compare the actual GDP with the model-predicted GDP. The research workflow proceeded as follows:

**1. Problem definition and hypothesis formulation:**

Clarifying the test objective as comparing distributional differences between groups, formulating statistical hypotheses.

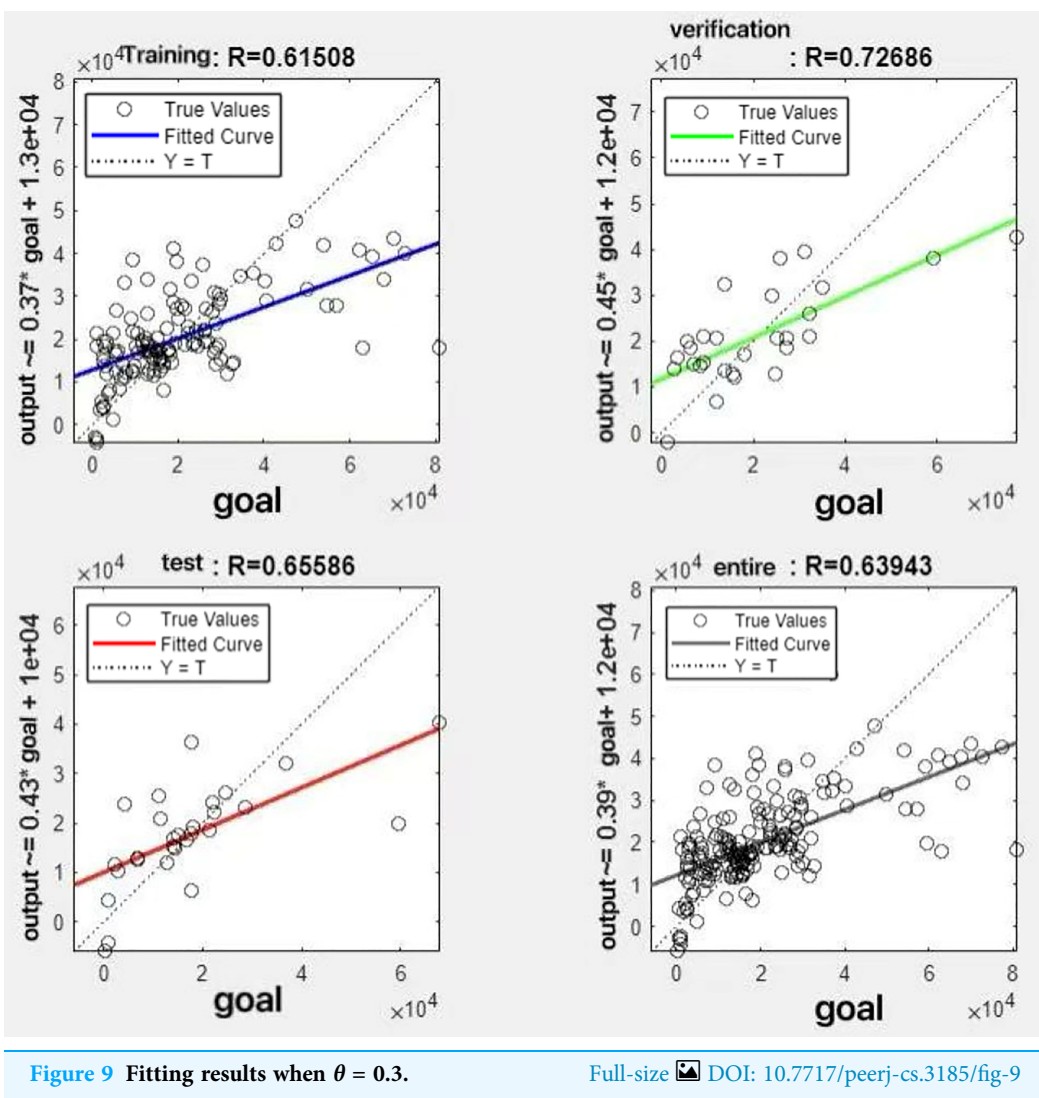

**Figure 9 Fitting results when $\theta$ = 0.3.**

Null hypothesis ($H_0$): both groups of data originate from the same distribution; alternative hypothesis ($H_1$): there exists a significant distributional difference (two-tailed test).

**2. Data verification and preprocessing:**
Examining data characteristics (small sample size, non-normal distribution), performing rank transformation on continuous variables, and handling ties using the average rank method.

**3. Test implementation and computation:**
Calculating inter-group differences based on the Mann-Whitney U statistic formula, estimating the difference distribution through Bootstrap resampling (1,000 repetitions) to compute a 95% confidence interval (percentile method), and deriving the $p$-value *via* the normal approximation method.

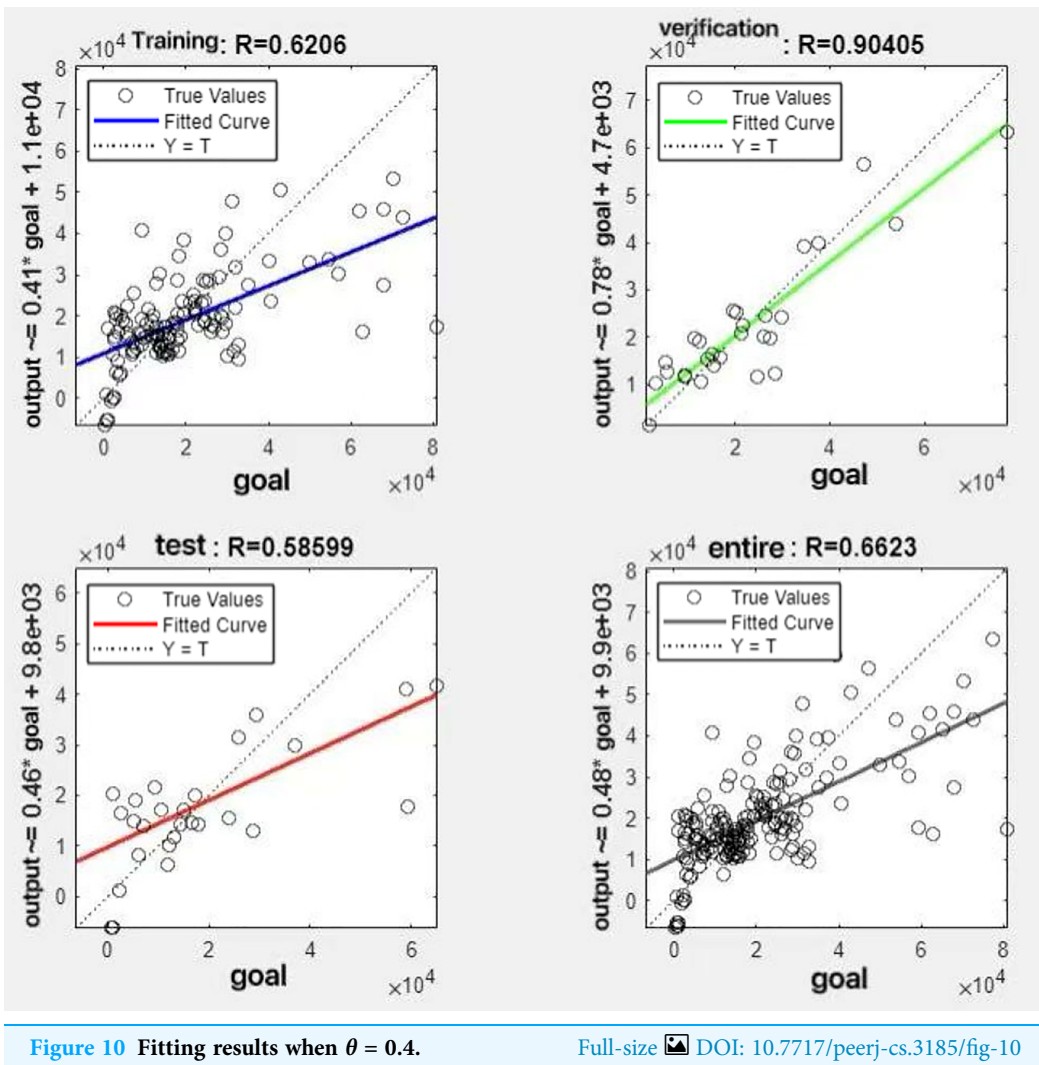

**Figure 10 Fitting results when $\theta = 0.4$.**

Results showed a test statistic Z = 2.035, an exact *p*-value of 0.668, and a Bootstrap confidence interval of [103.373–6,989.039], leading to rejection of the alternative hypothesis (H$_1$) and indicating that both datasets originate from the same distribution. Detailed results are presented in Table 3.

## Feature fusion parameter selection

In the R&P-NLPG model, data fusion employs a convex combination. A loop iterates through $\theta$ parameter values in increments of 0.1 (ranging from 0 to 0.9), completing nine iterations. The resulting nine $\theta$ values are then tested. We tested nine $\theta$ values to find the best one. The data was split 70% for training, 15% for testing, and 15% for validation. The optimal parameter $\theta$ was selected by comparing the R-values of models across data splits. Results are presented in Figs. 7 to 15.

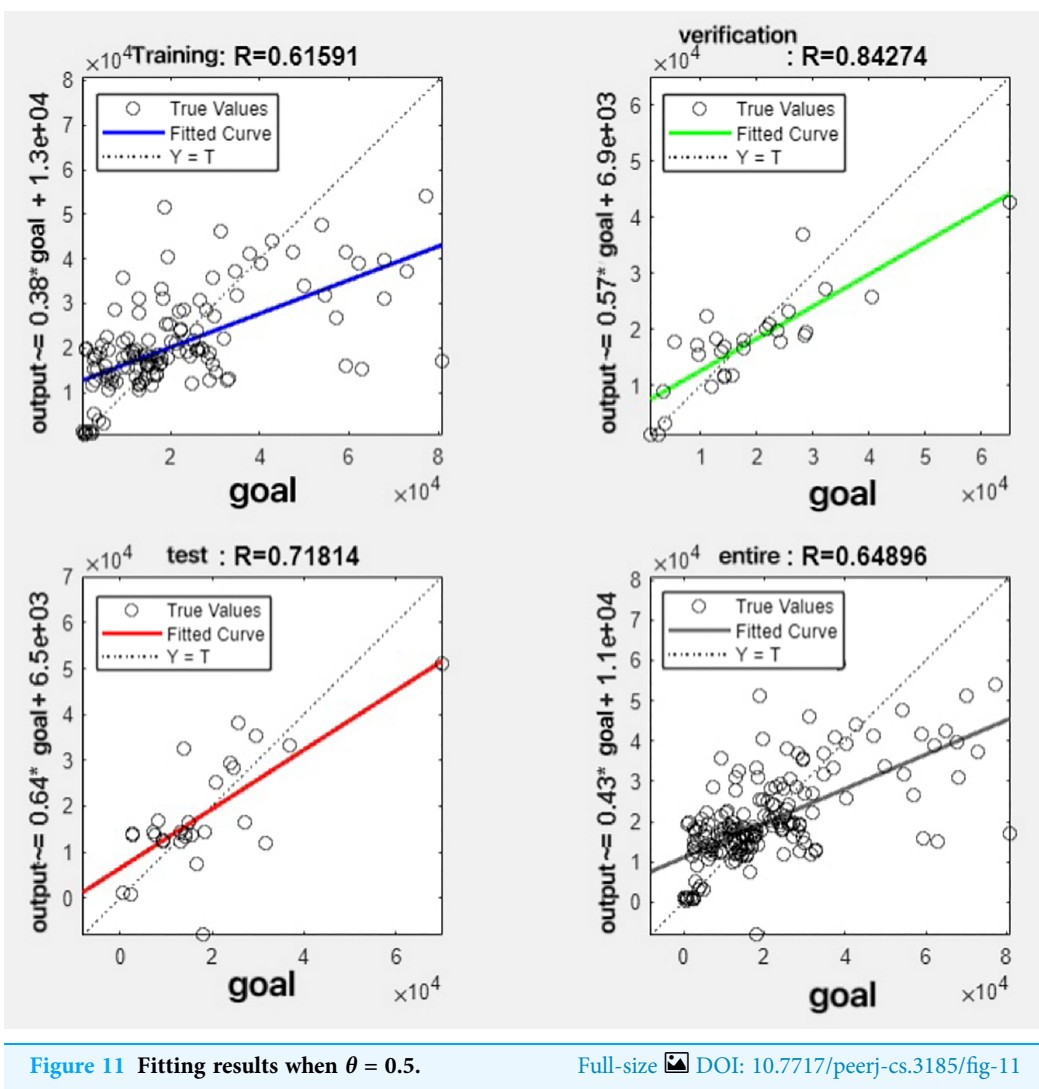

**Figure 11 Fitting results when θ = 0.5.**

It can be seen from these figures. When $\theta = 0.1$, the R-values of the validation and test sets are 0.79635 and 0.4519, respectively, which is shown in Fig. 7; when $\theta = 0.2$, they are respectively 0.7638 and 0.73128, which is shown in Fig. 8; when $\theta = 0.3$, they are respectively 0.72686 and 0.65586, which is shown in Fig. 9; when $\theta = 0.4$, they are respectively 0.90405 and 0.58599, which is shown in Fig. 10; when $\theta = 0.5$, they are respectively 0.84274 and 0.71814,which is shown in Fig. 11; when $\theta = 0.6$, they are respectively 0.84218 and 0.69989, which is shown in Fig. 12; when $\theta = 0.7$, they are respectively 0.9149 and 0.49333, which is shown in Fig. 13; when $\theta = 0.8$, they are respectively 0.60297 and 0.62862, which is shown in Fig. 14; when $\theta = 0.9$, they are respectively 0.71676 and 0.63206, which is shown in Fig. 15. From the above values, we can see that different $\theta$ values result in different fitting accuracy of the model in the validation and test sets. Among them, the model showed relatively good performance on

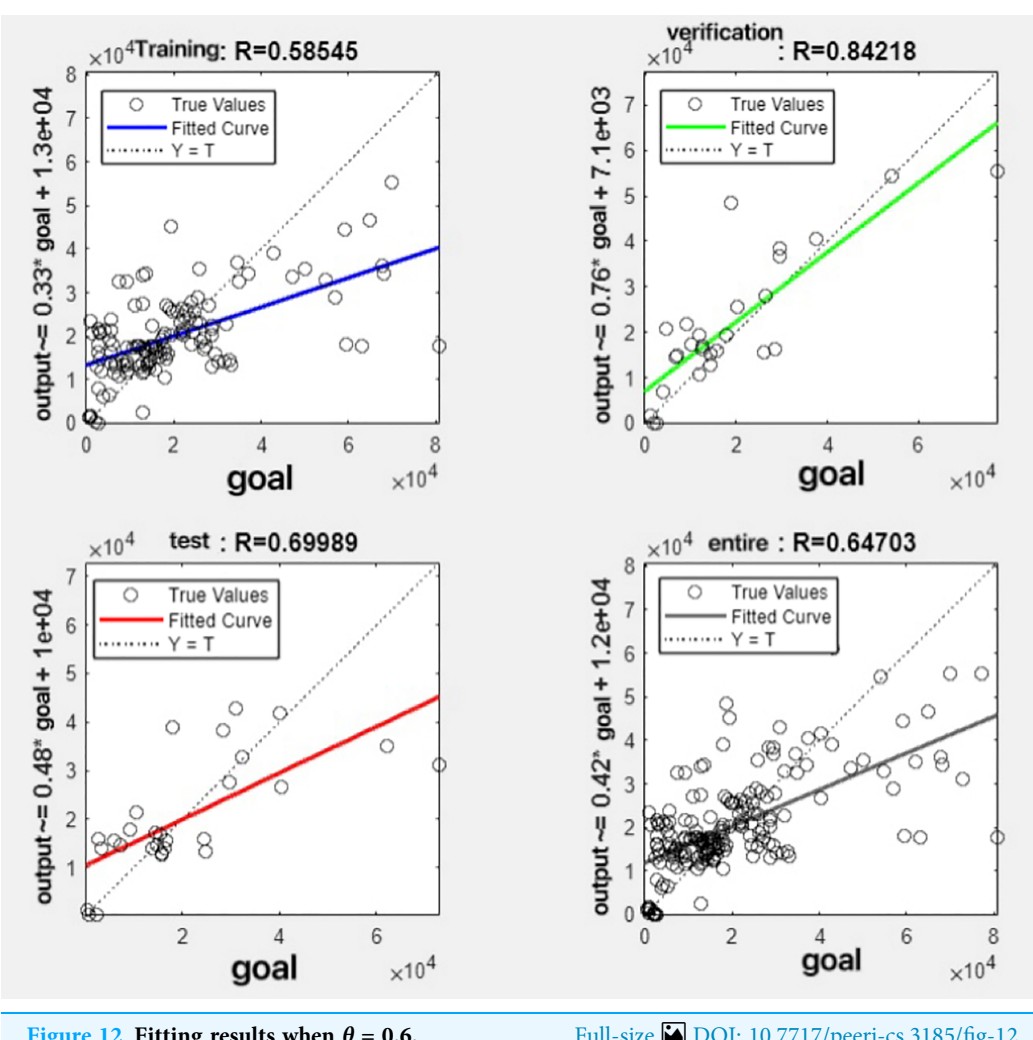

**Figure 12 Fitting results when θ = 0.6.**

both the test and validation sets when $\theta = 0.5$. Therefore, the optimal value of $\theta$ is determined to be 0.5.

## Model comparison

This experiment compared GDP predictions based on single variate model (nighttime light), existing multivariate model, and R&P-NLPG model. When using only single variate model (nighttime light), it can achieve an MSE of 0.0156 and an $R$-value of 0.5754. For existing multivariate models, the MSE was 0.0250 and the $R$-value was 0.5682. In contrast, the R&P-NLPG model yielded an MSE of 0.0147 and an $R$-value of 0.7488. The results demonstrate that R&P-NLPG model not only offers a novel approach but also enhances accuracy and fitting performance compared to single-variable model and multivariate model. Detailed comparisons are provided in Table 4.

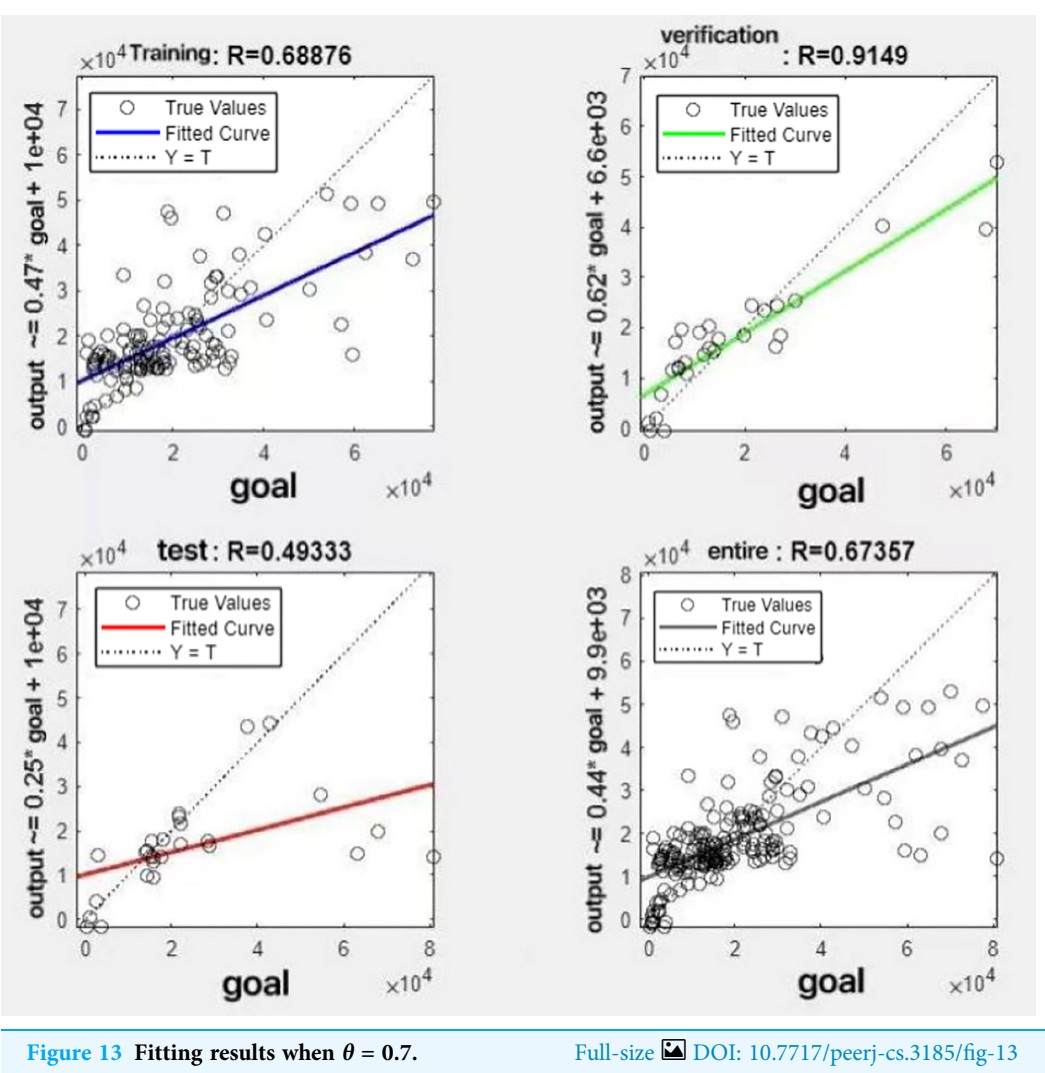

**Figure 13  Fitting results when $\theta$ = 0.7.**

## DISCUSSION

We discusses the findings from the Results section, explores their implications for the field of data prediction, discusses the practical significance of the model, and identifies its potential limitations.

We proposes a novel approach in the field of data prediction by employing convex combinations to integrate heterogeneous data types as feature inputs. The experiment shows that R&P-NLPG model has relatively good fitting performance and smaller prediction errors when compared with single-feature model and multi-feature method. The R-value of our model is 0.7488, and the prediction errors is between 0.006 and 0.008. Our model can achieve higher accuracy compared to traditional methods that rely solely on single-feature data or multi-feature data.These results indicate that the our model exhibits high reliability in predicting urban GDP.

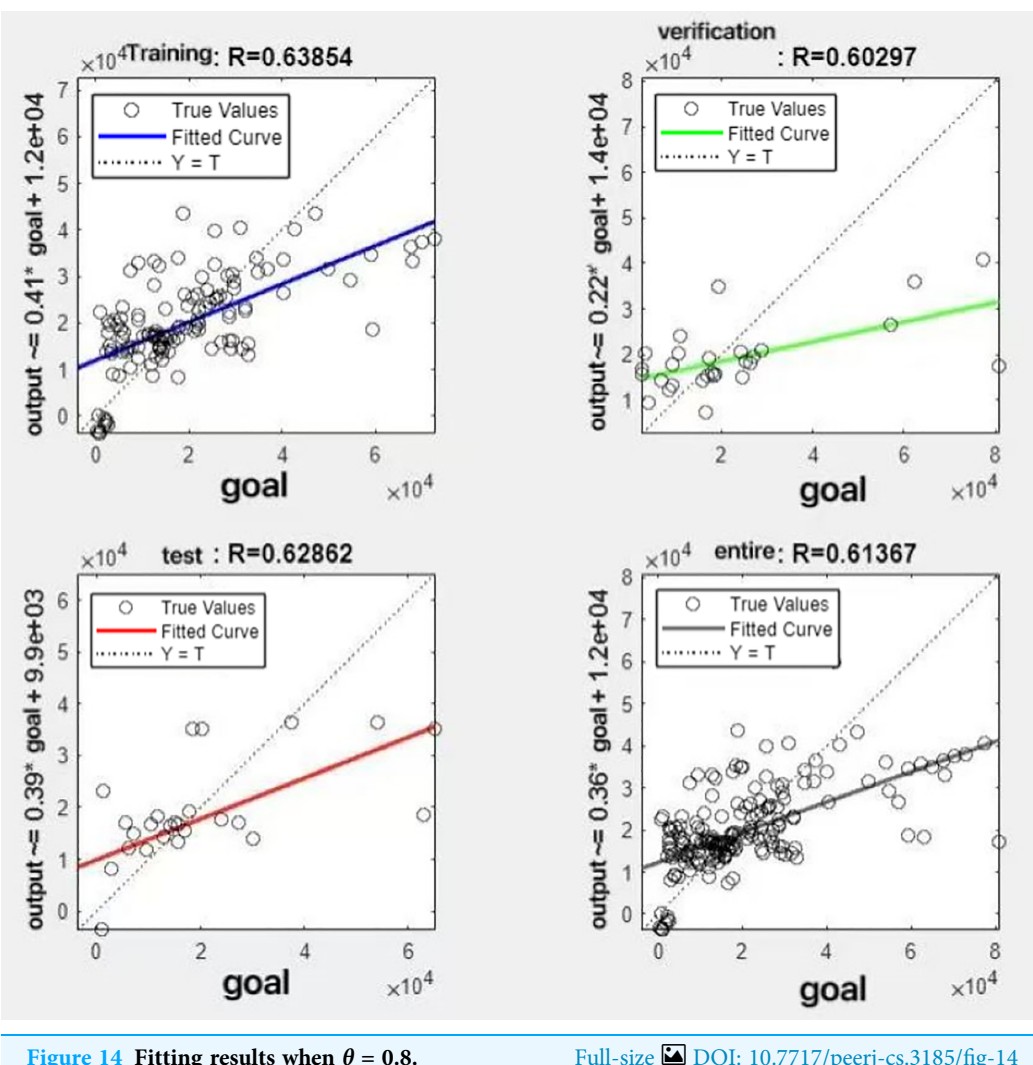

**Figure 14 Fitting results when $\theta$ = 0.8.**

This study employs Chinese cities as case examples; applying the model to predict urban areas in other countries would require parameter retraining to ensure accurate. The model can assist in urban development planning by identifying key development zones, predicting and regulating the interplay between industrial growth and air pollution, and promoting sustainable urban economic development. Furthermore, it offers evidence-based support for policy formulation. For example, when there is too much deviation between the predicted GDP and the government's expected economic goals, the government can formulate policies to promote preemptive intervention. For GDP, the model can provide guidance for the future direction of urban development. For nighttime lighting, the intensity of nighttime lights is strongly correlated with economic activities (*e.g.*, industrial output, energy consumption). Furthermore, the spatial coupling of nighttime light data with PM2.5 distributions enables the model to identify industrial clusters and trace pollution diffusion pathways. For PM2.5, it can improve the city's insight into air pollution and improve the city's air quality.

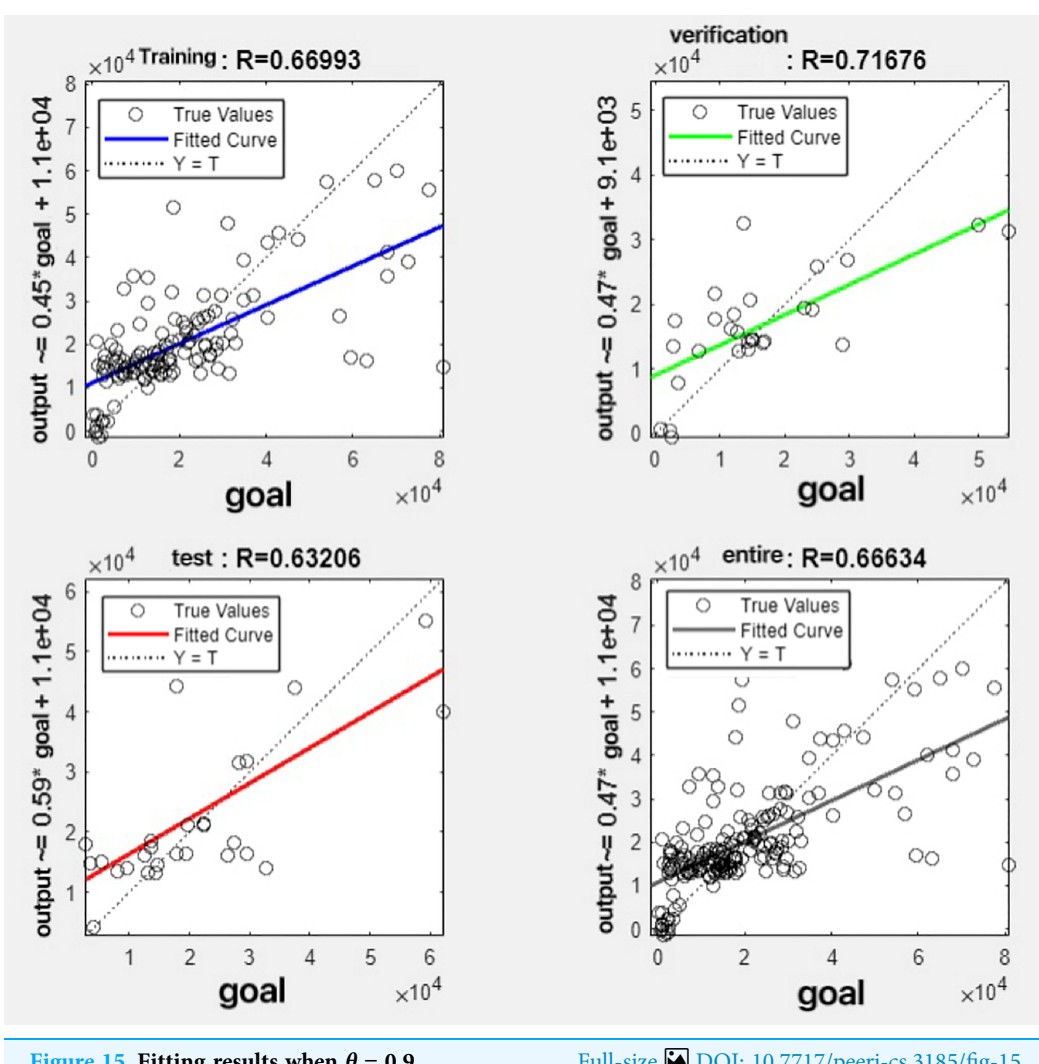

**Figure 15 Fitting results when $\theta$ = 0.9.**

**Table 4 Comparison of fitting results from different feature input methods.**

| Input variables | MSE | R |
|---|---|---|
| Fitting results with single nighttime lighting input | 0.0156 | 0.5754 |
| The fitting results of existing multivariate models. | 0.0250 | 0.5682 |
| Model fitting results of the R & P-NLPG | 0.0147 | 0.7488 |

While the model demonstrates corresponding strengths, it also has limitations. The primary limitation of the R&P-NLPG model lies in its heavy reliance on two input features: PM2.5 and nighttime light data. If either feature is missing, the model reverts to traditional single-feature prediction methods. Additionally, due to the need for multiple features in the R&P-NLPG model, processing speed is a weakness compared to models that use a single feature input. Future optimization efforts should prioritize developing rapid and high-precision techniques for feature acquisition. Nonetheless, the model retains robust

predictive accuracy, outperforming existing approaches under complete feature availability.

## CONCLUSIONS

To enhance the prediction accuracy of urban GDP models, this study proposes the R&P-NLPG model. Compared with traditional multi-feature input methods, its novelty lies in integrating PM2.5 data and nighttime light data together through convex combination methods as input. Leveraging a BP neural network fitting function, compared to traditional single-feature model and multi-feature input model for urban GDP prediction, our model can enhance predictive accuracy. After retraining the model by the corresponding data of different city or region, R&P-NLPG model can predict precise GDP at different development levels.

### Funding

This work was supported by the National Natural Science Foundation of China (No. 62262055); Qing Lan Project of Jiangsu Province; Suqian Talent Xiongying Project (No. 2023-0035); Science and Technology Foundation of Guizhou Province (No. [2019] 1447); Nature Science Foundation of Guizhou Educational Department (No. [2022]100); High Level Talent Foundation of Suqian University (No. 2024XRC011); Suqian Sci&Tech Program (No. L202309). There was no additional external funding received for this study. The funders had no role in study design, data collection and analysis, decision to publish, or preparation of the manuscript.

### Grant Disclosures

The following grant information was disclosed by the authors:
National Natural Science Foundation of China: 62262055.
Qing Lan Project of Jiangsu Province.
Suqian Talent Xiong ying Project: 2023-0035.
Science and Technology Foundation of Guizhou Province: [2019]1447.
Nature Science Foundation of Guizhou Educational Department: [2022]100.
High Level Talent Foundation of Suqian University: 2024XRC011.
Suqian Sci&Tech Program: L202309.

### Competing Interests

The authors declare that they have no competing interests.

### Author Contributions

- Sen Chen conceived and designed the experiments, performed the experiments, analyzed the data, performed the computation work, prepared figures and/or tables, authored or reviewed drafts of the article, and approved the final draft.

- Junke Li conceived and designed the experiments, performed the experiments, analyzed the data, prepared figures and/or tables, and approved the final draft.

## Data Availability

The data and code are available in the Supplemental Files.

The third-party data is available at the China Statistical Yearbook-National Bureau of Statistics: https://www.stats.gov.cn/sj/ndsj.

## Supplemental Information

Supplemental information for this article can be found online at http://dx.doi.org/10.7717/peerj-cs.3185#supplemental-information.

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
