# Peer review of "Research on the relationship and prediction model between nighttime lighting data, pm2.5 data, and urban GDP"

_PeerJ Computer Science, doi:10.7717/peerj-cs.3185_

## Round 0.1 · original submission · Minor Revisions

Dear Authors,

I carefully read the reviewers’ reports and concluded that this contribution is eligible for publication. However, some minor improvements are needed, as highlighted by the reviewers.

The main concern, raised by Reviewers 2 and 3, is the need to strengthen the statistical validation. The use of Pearson correlation alone may not be sufficient. Please consider integrating additional validation techniques, such as hypothesis testing, confidence intervals, sensitivity analysis, or cross-validation, to reinforce the robustness of your findings.

As noted by all reviewers, the manuscript is generally well-structured and written in professional English. However, several sections could benefit from more concise and simplified language. Please ensure that a generic reader can clearly understand the core ideas and findings of your work. Full proofreading for clarity and conciseness is recommended.

Further, please ensure that all preprocessing steps and model parameters (e.g., neural network architecture, hyperparameters, activation functions, training procedure) are detailed. This is essential to allow for full reproducibility and transparency. Reviewers also pointed out that the rationale for certain choices—such as the use of a convex combination for feature fusion—should be better explained and supported.

Regarding the literature review, while comprehensive, it would benefit from the inclusion of more recent references. Also, in several parts of the manuscript, key claims lack specific references (e.g., lines 48–50 and 60–65, as noted by Reviewer 3). Please revise these sections and ensure all statements are properly supported.

Some figures need to be revised for better clarity and resolution. Consider adding more descriptive captions, consistent color schemes, direct labels or error bars where appropriate, and clearer legends—especially for Figures 4 and 5.
Please also review the acronyms used in the manuscript. All acronyms (e.g., “PM”) must be defined upon first use. Alternatively, consider adding an acronym table at the beginning of the manuscript for clarity.

Finally, while your model shows promising results, the novelty in comparison to existing multi-variable models should be more explicitly discussed. Also, consider expanding the discussion on model limitations, generalizability, and the potential implications of your findings in urban planning and policy contexts, as suggested by Reviewer 3.

Reviewer 1 ·

Basic reporting

The manuscript is well-structured and written in clear, professional English, though some sentences are overly complex and could benefit from simplification. The literature review is comprehensive and includes relevant references. Nevertheless, the article would benefit from the inclusion of some more recent references. Figures and tables are appropriately labeled and useful in illustrating key points, though some images could be higher resolution for better readability. Overall, minor revisions are needed to improve readability and figure clarity.

Experimental design

The research question is well-defined and addresses a relevant gap in GDP prediction models by integrating multiple factors. The methodology is clearly presented, with detailed descriptions of data collection, preprocessing, and modeling. The use of a neural network and feature fusion is innovative, but some preprocessing steps and model parameter selections could be explained in greater detail. While the study is replicable, adding more details about tuning processes would improve clarity. The work meets high technical and ethical standards, with only minor refinements needed for greater transparency.

Validity of the findings

The findings are well-supported by robust statistical analysis, with a strong correlation between the variables confirming the study’s hypothesis. The proposed model demonstrates an improvement in GDP prediction accuracy compared to single-feature models, but there is still room for optimization. The conclusions align with the results and acknowledge potential areas for further research, particularly in incorporating additional influencing factors. While the study’s validity is sound, a brief discussion of model limitations and potential refinements would strengthen the argument.

Additional comments

The acronym "PM" is never defined in the article.

Reviewer 2 ·

Basic reporting

The paper is generally well-structured and provides a solid background on the relationship between nighttime lighting, PM2.5, and GDP. The writing is mostly clear and professional, but some sections could benefit from minor language refinements to improve readability. Certain sentences are slightly redundant or could be more concisely stated. A thorough proofreading for clarity and conciseness would enhance the overall presentation.

The literature review is comprehensive, citing relevant studies to justify the choice of data sources and methodology. However, while previous studies on nighttime lighting and PM2.5 in relation to GDP are discussed, the novelty of the proposed approach could be emphasized more clearly. Specifically, explaining how this model advances beyond existing methods would help better position its contribution. The manuscript is self-contained and presents its results in a logical manner. The hypotheses are addressed, and the conclusions are well-grounded in the findings. However, including a brief discussion on the potential limitations of the approach—such as data availability, model assumptions, or potential generalization issues—would add valuable depth to the analysis.

Overall, the paper meets the basic reporting standards but would benefit from minor refinements in clarity, conciseness, and highlighting its novel contributions.

Experimental design

The research presents an original and relevant approach to GDP prediction using nighttime lighting and PM2.5 data, which aligns well with the journal's scope. The research question is clearly defined, and the study effectively highlights the need for a multi-factor approach to improve prediction accuracy. However, while the paper emphasizes the gap in single-variable GDP prediction models, the contribution could be framed more explicitly by comparing it to other multi-factor approaches in the literature.

The methodology is well-detailed, with clear steps for data collection, preprocessing, correlation analysis, feature fusion, and neural network implementation. The description of data sources and processing steps ensures that the study can be replicated. However, additional details on the choice of hyperparameters, neural network architecture (e.g., number of layers, neurons, activation functions), and training process would provide better insight into the robustness of the model. A brief discussion on potential overfitting and how it was mitigated would also strengthen the methodological rigor. The choice of a convex combination for feature fusion is reasonable, but it would be helpful to provide a more detailed justification for why this method was selected over other possible fusion techniques. Additionally, while the experimental setup is well-structured, the validation approach could be expanded by comparing the proposed model’s performance with existing GDP prediction models rather than just single-variable baselines.

Overall, the experimental design is strong, with a well-defined methodology and reproducible steps. Clarifying certain methodological choices and expanding on validation and comparison with alternative models would further enhance the study’s credibility.

Validity of the findings

The study presents a novel approach to GDP prediction by integrating nighttime lighting and PM2.5 data, but its novelty could be emphasized more clearly in comparison to existing multi-variable models. While the paper effectively establishes the rationale for combining these factors, a more explicit discussion of how this approach advances prior research would better highlight its contribution.

The provided data appear robust, with clear preprocessing steps ensuring consistency. However, while Pearson correlation analysis is used to establish relationships between variables, additional statistical validation—such as confidence intervals, hypothesis testing, or cross-validation—could further support the model’s reliability. Additionally, while the neural network approach is well-described, including an assessment of model stability (e.g., sensitivity analysis or performance on an independent test set) would strengthen the claims. The conclusions are well-stated and logically follow from the findings, with clear links to the original research question. However, a brief discussion of the model's generalizability across different geographic or economic contexts would be valuable. Addressing potential limitations, such as data availability, economic anomalies, or regional variations in pollution and lighting, would provide a more balanced perspective.

Overall, the findings are valid and well-supported, but adding comparisons to existing models, additional statistical validation, and a discussion of broader applicability would enhance the study’s credibility.

Reviewer 3 ·

Basic reporting

The paper aims to develop a prediction model for urban GDP that integrates nighttime lighting data and PM2.5, two key indicators that in literature independently are correlated with economic activity, and with GDP data. By using data fusion approach and neural network for fitting function to improve model accuracy and better capture the multifaceted influences on urban economic development, the authors seeks to overcome the limitations of previous studies based on single input variables. To support their innovative contributions, the authors cite some studies that relate to both nighttime lighting and PM2.5 as predictors of urban GDP, identifying gaps that the R&P-NLPG model seeks to address. In so doing, sometimes they lack to indicate appropriate references to support some declaration, as happen for the sentence from line 48 to 50. The same consideration about the statement from line 60 to 65 and from 145 to 145. So, be careful above these shortcomings, because without appropriate indications, these statements could be considered personal opinions. Moreover, some works are mentioned in a general way (e.g., "Zhan Qiwen et al., 2021") without providing specific details about how these studies directly inform the methodology or findings of the current paper. It would be helpful to explicitly mention how these studies contributed to the development of the research question or the design of the paper’s prediction model. As more discussion could be provided on the methods used in previous research. For example, while the paper mentions studies using linear models or spatial regression models, it doesn't clearly explain why the proposed neural network model might offer advantages over these methods. Therefore, while the paper references relevant prior literature, it is possible to improve in how the references are integrated into the narrative. The connections between the cited studies and the research objectives could be made clearer.
The paper’s structure meets the journal's criteria. As requested it is organized into clearly defined sections. Nevertheless, clarifying certain technical sections it could enhance readability. For instance, simplifying the descriptions of data preprocessing, the neural network architecture, and the convex combination method would help readers who are less familiar with these techniques to be more interested. Same considerations about the figures whose resolution have to be improved. They effectively illustrate the model framework, data acquisition processes, and experimental results but adding more detailed captions or legends for some figures will allow for enhancing clarity. Some labels are too brief and do not fully explain what the figure shows or why it matters. For examples figures 4 and 5, show error metrics across different models or regions but could be hard to interpret at a glance if bars or points are not clearly labelled. A suggestion could be to use consistent colours and include a more descriptive legend and for more precise interpretation to add direct data labels or error bars.

Experimental design

By combining computational science techniques—such as neural networks, data fusion, and statistical analysis—with real-world applications, the research is well within the scope of the journal which welcomes studies applying computational methods to solve complex problems in various domains. The paper clearly identifies a gap in current methodologies and proposes a novel approach that integrates both nighttime lighting and PM2.5 data using data fusion and neural network techniques, addressessing an important aspect of urban economic analysis and offering significant potential for improving GDP prediction accuracy. Nevertheless, while the paper identifies the gap related to the use of single-variable models, it could further elaborate on why existing multi-variable approaches have not succeeded or been explored in depth. This would clarify the novelty of the approach and help emphasize why the current study's methodology is particularly valuable. More discussion could be provided about how integrating PM2.5 data with nighttime lighting data adds an innovative dimension to GDP prediction. Some references to other interdisciplinary studies that have attempted to combine environmental data with economic models could strengthen this point.
The paper details the data sources (nighttime lighting, PM2.5, and GDP statistics) and describes the general preprocessing steps. So overall, sufficient detail to understand the method, but more technical specificity is needed for exact replication. For example, the neural network architecture and training details (e.g., hyperparameters, number of layers, activation functions, optimizer settings) are not fully specified, making it difficult for others to replicate or evaluate the model’s design choices. The convex combination method for integrating nighttime light and PM2.5 data is innovative, but the process for determining the optimal weights (α and β) is not clearly detailed. It is unclear whether these weights are learned, empirically set, or optimized via validation. Moreover, the study does not discuss confidence intervals, sensitivity analysis, or error propagation, which would be useful to understand the robustness of predictions. Finally, because the study focuses on a specific region (likely in China, though the exact area is not always emphasized), this doesn’t allow for a generalization of the model to other cities or countries with different lighting, pollution, or economic patterns. About the data some shortcomings could be raised. The dataset used in the study is generally suitable for the research goal, but, in particular for GDP, the study uses data from a single year (2020), without considering other important factors that influence GDP, such as: population density, industrial composition, and Government policy and a sequence of data able to capture trends. This could lead to bias in the model, reducing its explanatory power. In this view, the paper is just an exercise to show the potential of the approach.

Validity of the findings

The Data available from the Dryad Digital Repository(Dryad), a well-respected discipline-specific open-access repository—satisfies standards of transparency and reproducibility.The discussions and conclusions appears just a summary of the research. The implications about the applications of the research in some areas are totally missing. The model’s ability to predict GDP more accurately by incorporating both nighttime lighting and PM2.5 data could have significant implications for urban planning, policy, and sustainable development. Therefore, a deepening of this question could overcome the idea that the research is just a technical exercise

Additional comments

The paper aims to develop a new prediction model—named the R&P-NLPG model—that integrates both nighttime lighting data and PM2.5 data to more accurately predict urban GDP. It seeks to overcome the limitations of previous studies that relied on single input variables by using a data fusion approach (via a convex combination of features) and a neural network fitting function to improve model accuracy and better capture the multifaceted influences on urban economic development. While the paper is comprehensive, clarifying the points highlighted in previous could enhance readability and practical application of the research

---

## Round 0.2 · accepted · Accept

Dear Authors,

Thank you for addressing all reviewers' concerns.

Reviewer 1 ·

Basic reporting

The authors have sufficiently addressed the reviewers concerns, clarifying acronyms and improving the overall writing. However, I cannot find the students' names in the authors list.

Experimental design

no comment

Validity of the findings

no comment

Reviewer 3 ·

Basic reporting

ok

Experimental design

ok

Validity of the findings

ok

Additional comments

ok